# Vitamin D Modified DSS-Induced Colitis in Mice via STING Signaling Pathway

**DOI:** 10.3390/biology14060715

**Published:** 2025-06-18

**Authors:** Zhihao Wu, Baohua Ma, Min Xiao, Qian Ren, Yanhua Shen, Zhengyu Zhou

**Affiliations:** Laboratory Animal Center, Suzhou Medical College, Soochow University, Suzhou 215123, China; zhwu@suda.edu.cn (Z.W.); ffpc_151501@sohu.com (B.M.); xiaomin06052022@163.com (M.X.); renqian@suda.edu.cn (Q.R.); shenyanhua@suda.edu.cn (Y.S.)

**Keywords:** 1,25(OH)_2_D_3_, DSS-induced colitis, STING pathway, gut microbiota, inflammatory bowel disease

## Abstract

Previous research has established a correlation between low levels of vitamin D and the severity of inflammatory bowel disease. Our study provides evidence that vitamin D supplementation significantly mitigated colitis induced by dextran sulfate sodium through the suppression of dysregulated immune responses, reduction in inflammation, and restoration of gut microbiota equilibrium. Mechanistically, vitamin D exerts its effects by inhibiting the activation of the GATA1/STING signaling pathway, which leads to a downregulation of type I interferon production and the release of inflammatory cytokines. These actions are instrumental in maintaining the integrity of the intestinal barrier and alleviating gut dysbiosis, a characteristic feature of IBD. Vitamin D not only reduces pro-inflammatory cytokines but also promotes the proliferation of beneficial microbial communities, underscoring its dual role in immune regulation and microbiota homeostasis. In general, the modulation of the STING pathway by vitamin D indicates a potential novel therapeutic approach for the management of IBD.

## 1. Introduction

Inflammatory bowel disease (IBD) has been receiving increasing attention due to its rising incidence worldwide. IBD includes Crohn disease (CD) and ulcerative colitis (UC), which are characterized by persistent excessive inflammatory response, leading to symptoms such as rectal bleeding, intestinal stenosis, recurrent abdominal pain, and weight loss [1]. The pathogenesis of IBD involves multiple factors, such as various immune cells and inflammatory responses, intestinal microbiota, patient genetic factors, and environmental changes [2].

Vitamin D is derived from diet or synthesized by the skin and is hydroxylated in the liver and kidneys to form biologically active 25(OH)D_3_ and fully active 1,25(OH)_2_D_3_ [3,4]. The classic function of vitamin D is to regulate the homeostasis of calcium and phosphorus and govern the process of bone remodeling. Many studies have suggested a correlation between low levels of vitamin D and the onset of IBD. Vitamin D deficiency is highly prevalent among IBD patients, and there is a significant association between reduced serum vitamin D levels and disease activity [5]. In vivo, the biological effects of vitamin D are mediated through its binding to vitamin D receptor (VDR). The VDR is extensively expressed in intestinal epithelial cells, and it plays a crucial role in maintaining the integrity of the mucus layer and intestinal epithelium through various mechanisms, including regulation of tight junctions and adherens junctions, as well as secretion of antimicrobial peptides, such as cathelicidin and β2-defensins [6,7]. It has been shown that the lack of intestinal VDR leads to changes in the composition of gut microbiota in mice [8,9], and supplementation of vitamin D with the biologically active form of calcitriol (1,25(OH)_2_D_3_) can stimulate macrophages to produce antimicrobial peptides, thereby promoting bacterial eradication [10]. Furthermore, VDR is expressed by various immune cells, including macrophages, dendritic cells, B cells, and T cells [11,12], and vitamin D plays a pivotal role as a crucial regulatory factor in the innate immune response to pathogenic microorganisms and is extensively involved in immune modulation [13]. Consequently, vitamin D signaling plays a pivotal role in the pathogenesis of intestinal injury.

Stimulator of interferon genes (STING) serves as a pivotal signaling molecule in the cyclic GMP–AMP synthase (cGAS) pathway, enabling the recognition of microbial DNA, mitochondrial DNA, and mislocalized self-DNA within the cytoplasmic compartment. When cGAS binds to dsDNA, leads to the production of the second messenger 2′3′ cyclic GMP-AMP (cGAMP), cGAMP binds STING and resulting in STING dimerization, then interacts with TANK-binding kinase (TBK1) and activates its autophosphorylation [14]. This activation triggers downstream signaling cascades involving interferon regulatory factor 3 (IRF3) and nuclear factor-κB(NF-κB)/p65, ultimately facilitating the secretion of interferons, inducing type I IFN response and initiation inflammatory response [15]. The STING signaling pathway is a crucial nexus of immune and cellular biology, implicated in the pathogenesis of numerous diseases, including infections, autoimmune disorders, and malignant tumors [16]. In the context of IBD, damage to the intestinal mucosa results in cell death and the release of DNA. Consequently, the cGAS-STING signaling pathway may play a significant role in the pathogenesis and progression of IBD. This study aimed to investigate the potential role of vitamin D in DSS-induced colitis, thereby providing novel insights into the clinical management of intestinal diseases.

## 2. Materials and Methods

### 2.1. Animals

Male C57BL/6 mice, aged 6 weeks, were procured from Changzhou Cavens laboratory animal Co., Ltd. (Changzhou, China) and housed in the specific pathogen-free (SPF) animal facility at the Laboratory Animal Center of Suzhou Medical College, Soochow University (license number SYXK (Su) 2021-0065). The environmental conditions were as follows: a temperature of 22 °C ± 2 °C, humidity of 50% ± 5%, and a 12 h light–dark cycle, with ad libitum access to food and water. This experiment received approval from the Ethics Review Committee of the Experimental Animal Center at Suzhou Medical College, Soochow University (Approval No: 202309A0825).

### 2.2. Modeling and Evaluation of DSS-Induced Colitis

After 1 week of adaptive feeding, the mice were randomly allocated into four groups with equal sample sizes (*n* = 5). The control group (Con) and the vitamin D treatment group (Con+VD) were allowed to freely consume sterilized water for 6 weeks. The DSS-induced colitis group (DSS) and the vitamin D treatment colitis group (DSS+VD) were allowed to freely consume 2% DSS for 5 days, followed by a cycle of 10 days on sterile water (Figure 1A). This cycle of 2% DSS–sterile water was repeated three times [17]. The Con+VD group and the DSS+VD group received intraperitoneal injections of 1,25(OH)_2_D_3_ at a dosage of 5 μg/kg body weight (dissolved in peanut oil), administered once every 5 days. The control group and the DSS group were intraperitoneally injected with peanut oil as a control. All animals underwent weekly weight measurements, and a record of the disease activity index (DAI) was maintained, which includes 3 indicators: weight loss (0–4, 0 to 15% loss), stool consistency (0 points for normal, 2 points for loose stool, and 4 points for diarrhea), and hematochezia (0 points for normal, 2 points for positive occult blood, and 4 points for overt hemorrhage) [18]. After three cycles of DSS–sterile treatment, all animals were euthanized using CO_2_, and their colons were collected for measurement of length and photography. The colon tissue was fixed, dehydrated, and embedded in paraffin to make sections for staining, while the remaining tissues were frozen and stored at −80 °C. Blood was collected from saphenous vein, and after centrifugation at 3500× *g* for 15 min, the serum was collected for measuring total superoxide dismutase (SOD) and interferon-beta (IFN-β) activity in blood.

### 2.3. Real-Time PCR

Total RNA was extracted from the intestinal tissues using Triquick Reagent (Trizol Substitute, Solarbio, Beijing, China), followed by cDNA synthesis using a reverse-transcription kit. Primers (*IL-6*: F: gagaggagacttcacagagg, R: gtactccagaagaccagagg; *Il-1β*: F: gcaactgttcctgaactcaact, R: atcttttggggtccgtcaact; *Il-36*: F: gcagcatcaccttcgcttaga, R: cagatattggcatgggagcaag; *Il-23*: F: cagcagctctctcggaatctc, R: tggatacggggcacattattttt; *Il-17*: F: gaaggccctcagactacctcaa, R: tcatgtggtggtccagctttc; *Il-22*: F: tcgccttgatctctccactc, R: gctcagctcctgtcacatca; *STING1*: F: ggtcaccgctccaaatatgtag, R: cagtagtccaagttcgtgcga; *Ifn-β*: F: agctccaagaaaggacgaaca, R: gccctgtaggtgaggttgat; *Nrf2*: F: tagatgaccatgagtcgcttgc, R: gccaaacttgctccatgtcc; *Keap1*: F: cggggacgcagtgatgtatg, R: tgtgtagctgaaggttcggtta; *Mucin2*: F: agggctcggaactccagaaa, R: ccagggaatcggtagacatcg) were synthesized by Shanghai Shenggong Biotech Co., Ltd. using the Dr. oligo192 primer synthesizer (Biolytic China, Kunshan, China). All samples were run in triplicate and the concentration of cDNA was 100 ng/mL. Amplification was performed in an RT-PCR system with ChamQ SYBR qPCR Master Mix (Without ROX) and specific primers, under the following conditions: the protocol involved initial denaturation at 94 °C for 1 min, followed by annealing at 60 °C for 15 s and extension at 72 °C for 6 s. This amplification cycle was repeated for a total of 40 times. The expression levels of glyceraldehyde-3-phosphate dehydrogenase (GAPDH) were utilized to determine the normalized cycle threshold (Ct value), with GAPDH serving as a reference gene to assess the relative expression of other genes.

### 2.4. Histopathology

The colon tissue was fixed, dehydrated, embedded in paraffin, and sectioned into 5 µm thick slices for morphological observation and evaluation following hematoxylin–eosin staining. Alcian Blue Stain Kit (G1560, Solarbio, Beijing, China) was utilizes to perform staining of acidic mucins within colon tissue samples. The scoring of the tissue sections was performed by a registered veterinary pathologist who remained blinded to the sample grouping. Colonic histological activity index (HAI) evaluation was conducted based on indicators such as inflammation, edema, epithelial defects, crypt atrophy, hyperplasia, and dysplasia. The evaluation was graded on a scale of 0–4 [19].

### 2.5. Immunohistochemical Staining

The mouse colon tissue embedded in paraffin was cut into 5 µm thick sections and subjected to antigen retrieval using citrate salts, followed by incubation with antibodies against ZO-1 (1:500, Abcam, Cambridge, UK), GATA-1(1:5000, Abcam, Cambridge, UK), STING (1:2000, Abcam, Cambridge, UK), IRF3 (1:200, Abclonal, Wuhan, China), and IκBα (1:200, Abclonal, Wuhan, China) at 4 °C for 16 h. We used 3,3′-diaminobenzidine tetrahydrochloride (DAB) for color development, and hematoxylin was used for counterstaining for 1 min. Finally, the slices were examined under an optical microscope.

### 2.6. ELISA

Serum levels of SOD and IFN-β were quantified using specific assay kits (Solarbio, Beijing, China), and the absorbance values were measured with a microplate reader following the manufacturer’s instructions.

### 2.7. Analysis of Gut Microbiota

Microbial diversity analysis was commissioned to Guhe Information Technology Co., Ltd. (Hangzhou, China). The methods and complete 16s analysis results are included in the Appendix A.

### 2.8. Immunoblotting

Total protein was extracted from the colon segments at the same location. Cytoplasmic proteins were prepared using a cytoplasmic protein extraction kit (Beyotime, Shanghai, China). Each sample was loaded with 20 µg protein onto a 10% sodium dodecyl sulfate–polyacrylamide gel electrophoresis (SDS-PAGE) column and separated by electrophoresis before being transferred to a polyvinylidene fluoride (PVDF) membrane. The samples were incubated with antibodies against GATA-1(1:1000, Abcam, Cambridge, UK), STING (1:1000, Abcam, Cambridge, UK), IRF3 (1:1000, Abclonal, Wuhan, China), and IKBα (1:5000, Abclonal, Wuhan, China) at 4 °C for 16 h. Subsequently, the membranes were incubated with a secondary antibody coupled with horseradish peroxidase for 1 h, and the protein was revealed by Amersham ECL Select Western blotting reagent.

### 2.9. Statistical Analysis

The data analysis was performed using GraphPad software 8.0.2 (263). Parametric data were analyzed with one-way ANOVA and Tukey HSD for the post hoc, while nonparametric data were analyzed with the Kruskal–Wallis H. A significance level of *p* < 0.05 was considered statistically significant.

## 3. Results

### 3.1. Vitamin D Treatment Ameliorates DSS-Induced Colitis

Compared with the control group, the DSS group exhibited weight loss and colon shortening (Figure 1B,C). However, following vitamin D intervention, there was an improvement in weight loss and colon shortening (*p* < 0.05), as well as a reduction in DAI score, including weight loss, stool consistency, bleeding, and mortality (Figure 1B,C). Histopathological assessment showed that the DSS group exhibited extensive lymphocytic infiltration of the mucosal and submucosal layers, leading to the loss of normal crypt structure (Figure 1D). Severe defects were observed in intestinal glands, while other areas showed reduced crypts. Goblet cells in the intestinal mucosa disappeared, and there was an increase in submucosal proliferation. The DSS+VD group exhibited reduced lymphocytic infiltration and damaged area, as well as alleviated damage to crypts and goblet cells, with a decrease in submucosal hyperplasia (Figure 1D). The control group and the Con+VD group showed no histopathological changes, indicating that vitamin D intervention reduced tissue damage severity and improved symptoms of DSS-induced colitis (Figure 1E).

Nrf-2 and Keap1 are recognized as essential regulators of oxidative stress, offering cytoprotective effects by mitigating oxidative stress and inflammatory responses [20], which can indirectly serve as indicators of oxidative stress levels. Compared with the control group, the DSS group exhibited significantly increased transcription levels of Nrf-2 and Keap1, whereas vitamin D intervention effectively reduced their transcription levels (*p* < 0.005), indicating a decrease in oxidative stress levels (Figure 1F). The serum total SOD levels showed a similar trend across all groups (Figure 1F). The intestinal mucus layer is composed of approximately 30 fundamental proteins, which include mucins, antimicrobial peptides, and secreted immunoglobulin A, Mucin-2 is recognized as the most critical component [21]. Our study revealed that vitamin D significantly boosted Mucin-2 transcription in both normal and colitis model mice. Furthermore, we evaluated mucus secretion function and tight junction protein expression. Alcian blue staining can visually display the secretion function of intestinal glands. Compared with the control group and the Con+VD group, the DSS group showed extensive intestinal gland defects, while the degree of damage was significantly reduced in the DSS+VD group (Figure 1D). The downregulation of ZO-1 leads to increased permeability of tight junctions, and ZO-1 is crucial for the Wnt signaling pathway and mitotic spindle orientation, which makes ZO-1 essential for mucosal repair [22]. The immunohistochemical staining of tight junction protein ZO-1 showed a significant reduction in the DSS group compared with the control group, and there was a decrease in the submucosal layer and almost complete disappearance in the mucosal lamina propria. However, after vitamin D intervention, there was a significant increase in ZO-1 expression (Figure 1D).

### 3.2. Vitamin D Ameliorated Changes in Gut Microbiota in Mice with DSS-Induced Colitis

To explore the impact of vitamin D on gut microbiota, we examined the gut microbiota in mouse fecal samples using 16S sequencing. The sequencing data underwent splicing, quality control, and filtering processes to obtain an ASV feature table. The Shannon index curve demonstrated that each sample received ample sequencing data (Figure 2A). The Venn diagram analysis showed 844 overlapping operational taxonomic units (OTUs) among the four groups, assuming a sequence similarity of 97% (Figure 2B). The DSS group showed a decrease in both OTU abundance and evenness compared with the control group. However, after vitamin D intervention, there was a slight increase in OTU abundance and species diversity compared with the DSS group (Figure 2C). The gut microbiota composition was analyzed after vitamin D intervention. The genus level (Figure 2D) and heat map (Figure 2E) analysis revealed differences in gut microbiota composition between the DSS group and the other three groups. The Bray–Curtis method was used to analyze intergroup distribution and differences in PCoA for β diversity analysis. Significant differences were observed between PC1 and PC2, PC1 and PC3, and PC2 and PC3 (Figure 2F). The samples within the DSS group showed greater dispersion and deviation compared with the samples from other groups. Those results suggest that vitamin D intervention can improve gut microbiota alterations in the DSS-induced colitis.

### 3.3. Vitamin D Intervention Decreases the Expression of Inflammatory Factors

To investigate the impact of vitamin D on inflammation, fluorescent dyes were employed for quantitative PCR analysis of the transcriptional levels of various inflammation-related genes. Compared with the control group, vitamin D intervention did not alter the transcription levels of inflammatory factors. The expression levels of inflammatory factors such as IL-23, IL-1β, IFN-γ, IL-6, and IL-17 were significantly elevated in the DSS group (Figure 3A). However, following vitamin D intervention, the transcription of various inflammation-related cytokines in the DSS-induced colitis model was markedly suppressed and remained similar to that of the control group and the VD control group. In addition, vitamin D intervention led to a reduction in the transcription levels of STING and downstream IFN-β (Figure 3B,C).

### 3.4. Vitamin D Intervention Inhibits Activation of the STING Pathway

IκBα serves as a crucial signaling molecule within the NF-κB signaling pathway and is significantly associated with the inflammatory response. Immunohistochemical staining revealed a pronounced increase in positive expression of STING and downstream molecules IRF-3 and IκBα in the DSS group, whereas vitamin D intervention effectively attenuated the expression levels (Figure 4A). WB results showed that vitamin D intervention reduced the protein levels of STING, IRF3, and IKBα in DSS-induced colitis (Figure 4B). GATA1 is a member of the GATA transcription factor family, which is crucial for the functioning of the hematopoietic system and also exerts influence in other biological systems [23]. Furthermore, research has shown that GATA-1 can induce STING expression by promoting its transcription [24,25]. Immunohistochemical staining found that vitamin D intervention reduced the expression of GATA-1, suggesting that vitamin D may inhibit the activation of the STING pathway through GATA-1 and improve DSS-induced colitis.

## 4. Discussion

Our data demonstrate that the vitamin D intervention alleviated symptoms of DSS-induced colitis, reduced inflammation, preserved intestinal barrier integrity, and improved the composition of gut microbiota. Furthermore, the vitamin D intervention may inhibit the activation of the STING pathway through regulating GATA-1, then suppressing both the type I interferon response and inflammatory reactions.

Vitamin D plays a crucial role in a wide range of biological functions, such as regulating autophagy, cell proliferation, maintaining intestinal barrier function, modulating gut microbiota composition, and influencing immune responses [26]. In some patients with IBD, particularly those with CD, low vitamin D levels are associated with increased mucosal inflammation. Low vitamin D levels can serve as a biomarker for disease activity and a predictive factor for adverse prognosis in IBD patients [27]. Although numerous studies have identified vitamin D levels in patients as a potential marker for disease activity, our findings, along with other investigations involving DSS colitis models and clinical research, suggest that vitamin D intervention has a positive impact on IBD [28,29,30].

Previous studies have demonstrated the regulatory role of vitamin D in intestinal barrier [31,32,33]. Our findings are aligned with those previous reports—vitamin D exerts an inhibitory effect on the transcription of cytokines associated with inflammation while simultaneously enhancing the transcription of specific cytokines that contribute to the preservation of intestinal barrier integrity. For example, IL-22 plays a crucial role in host defense and tissue repair in the intestinal mucosa, it also exhibits proinflammatory characteristics, and proinflammatory cytokines such as IL-17 enhance its inflammatory effects [34,35]. Our study demonstrated that the transcription levels of both IL-17 and IL-22 were suppressed by vitamin D, which may prevent the proinflammatory effects of the synergy between IL-17 and IL-22 in DSS colitis.

The gut microbiota constitutes a diverse assemblage of microorganisms residing within the digestive tracts of humans, engaging in a symbiotic relationship with the host. Dysbiosis of the gut microbiota, which is defined by an imbalance between beneficial and opportunistic microbial populations, has been linked to IBD [36].The gut microbiota and the host participate in various mechanisms of reciprocal regulation, one critical mechanism is the cGAS/STING pathway, which plays a vital role in the recognition of exogenous DNA, including bacterial double-stranded DNA; this recognition subsequently activates the innate immune defense response, leading to the activation of the NF-κB pathway and regulation of the secretion of type I interferons [37]. The type I IFN response is essential for intestinal defense and homeostasis, and its dysregulation is associated with immune-mediated diseases such as IBD [38,39]. Research in other fields has suggested that vitamin D signaling modulates the activation of the STING/IFN pathway [40,41]. In a study on oral lichen planus, vitamin D has been shown to limit the activation of the STING signaling pathway by inhibiting the expression of GATA-1 [24]; this is consistent with our research. Several studies have indicated that GATA1 expression can be stimulated by HIF-1α signaling [42,43], whereas vitamin D appears to exert an inhibitory influence on the HIF-1α signaling pathway [44,45]. We propose the hypothesis that vitamin D may ameliorate the DSS-induced colitis by suppressing the HIF-1α/GATA1/STING signaling pathway.

Although our study has revealed several discoveries, it also has limitations. First, we have not extensively researched the exact mechanism by which vitamin D affects the GATA-1/STING/IFN pathway. Second, the interaction between vitamin D, the host, and gut microbiota in the context of DSS-induced colitis has not been thoroughly explored. We found that vitamin D can improve the disrupted gut microbiota in DSS-induced colitis; however, its mechanism is not yet fully understood. Additionally, the interaction between gut microbiota and the immune system in DSS-induced colitis has not been comprehensively investigated, particularly under vitamin D intervention. Gut microbiota regulates immune responses through the cGAS/STING pathway, impacting cell survival and proliferation, including processes such as apoptosis, necroptosis, and pyroptosis. Future studies should explore these aspects to establish a potential theoretical framework for IBD treatment. However, it is important to exercise caution when interpreting and applying these findings due to differences between animal models and human patients. Furthermore, there are limitations in our research; due to technical constraints, we were unable to employ the FITC-dextran method for evaluating intestinal barrier integrity, a technique recognized for its significant value in intestinal barrier integrity research. Future investigations should incorporate this method to yield more comprehensive data. Nevertheless, our results suggest novel therapeutic potential for vitamin D in ameliorating colitis through immune response inhibition, protection of intestinal barrier function, and regulation of gut microbiota composition.

## 5. Conclusions

Vitamin D intervention suppressed the GATA1/STING signaling pathway and both type I interferon response and inflammatory reaction, thereby preventing abnormal immune reactions that lead to dysbiosis of the gut microbiota and gastrointestinal diseases.

## Figures and Tables

**Figure 1 biology-14-00715-f001:**
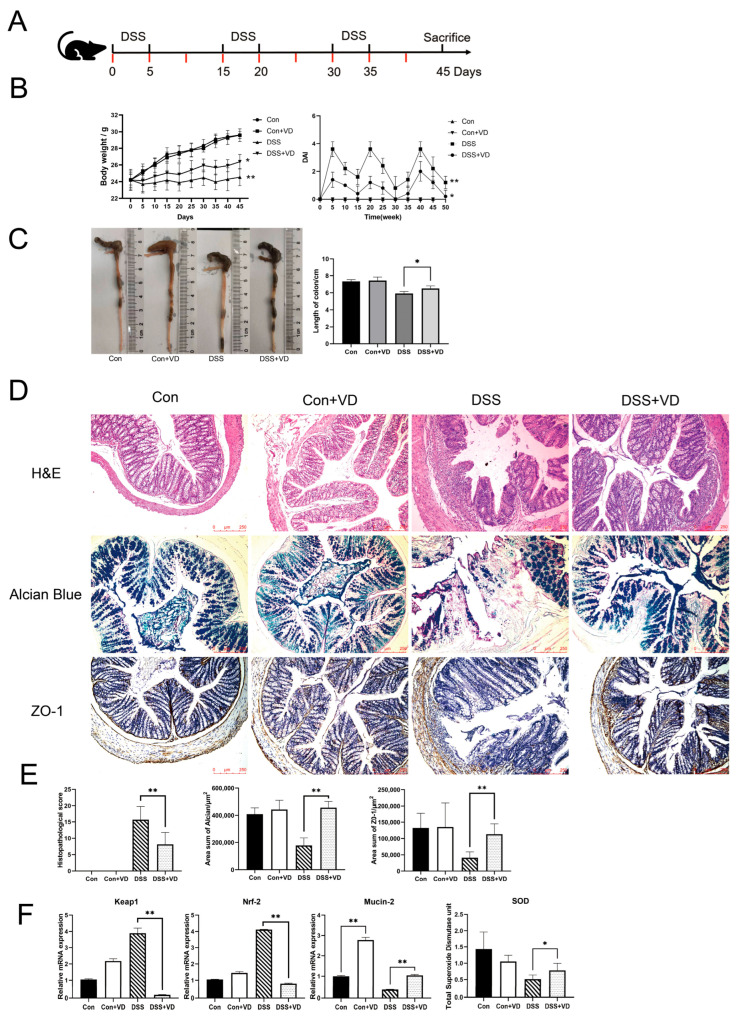
Modeling and evaluation of DSS-induced colitis. (**A**) Schedule of DSS-induced chronic colitis, the red mark indicates the schedule for 1,25(OH)_2_D_3_ injection. (**B**) Mice body weight and DAI score. (**C**) Changes in colonic length. (**D**) Hematoxylin–eosin staining, Alcian blue staining and immunohistochemical staining of ZO-1(magnification, ×100). (**E**) Histopathological score and positive staining area statistics of alcian blue and ZO-1. (**F**) Relative mRNA levels of colonic *Keap1*, *Nrf-2*, and *Mucin-2*, and the serum total SOD levels. Data are expressed as the means ± SD (*n* = 5). * *p* < 0.05 and ** *p* < 0.01. Each group contained five mice.

**Figure 2 biology-14-00715-f002:**
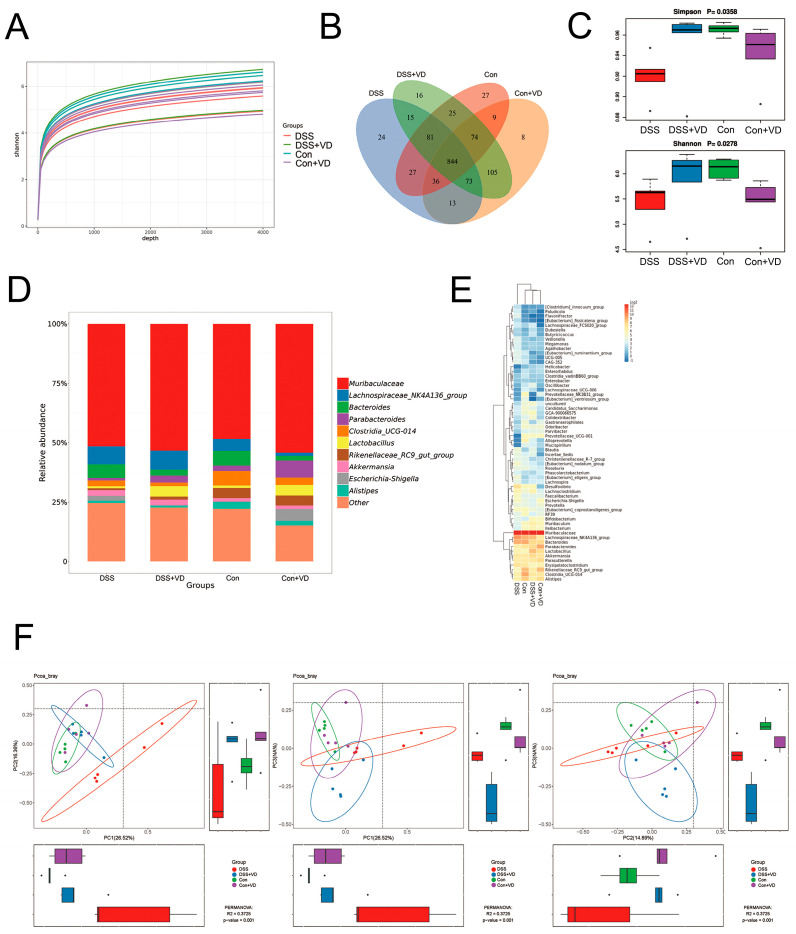
Vitamin D ameliorated changes in gut microbiota in the mice with DSS-induced colitis. (**A**) Shannon index curve. (**B**) The Venn diagram analysis. (**C**) α diversity analysis with Simpson and Shannon indexes. (**D**) Relative abundance in genus level. (**E**) Heat map of relative abundance in genus level. (**F**) β diversity analysis with PCoA.

**Figure 3 biology-14-00715-f003:**
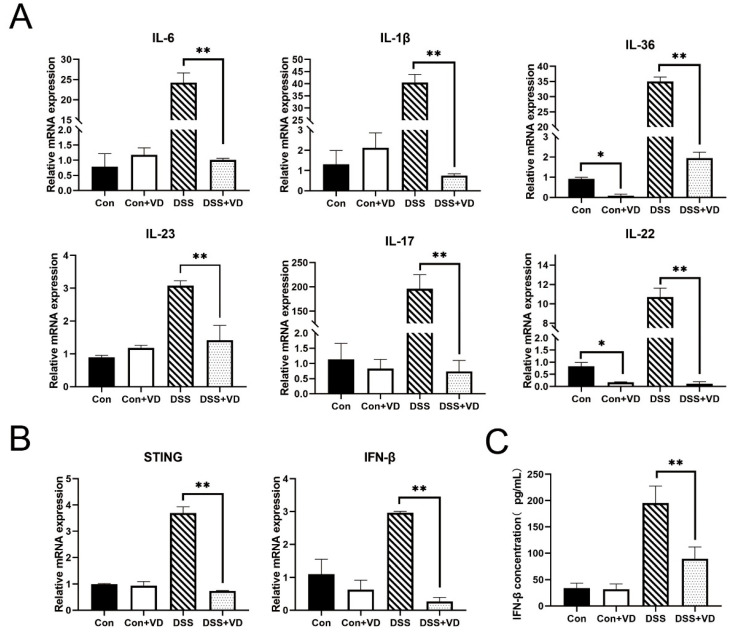
VD treatment inhibits inflammatory response. (**A**) Relative mRNA levels of colonic inflammatory cytokines *IL-6*, *IL-1β*, *IL-36*, *IL-23*, *IL-17* and *IL-22*. (**B**) Relative mRNA levels of colonic STING-related cytokines *STING* and *IFN-β.* (**C**) The serum *IFN-β* concentration. Data are expressed as the means ± SD (*n* = 5). * *p* < 0.05 and ** *p* < 0.01. Each group contained five mice.

**Figure 4 biology-14-00715-f004:**
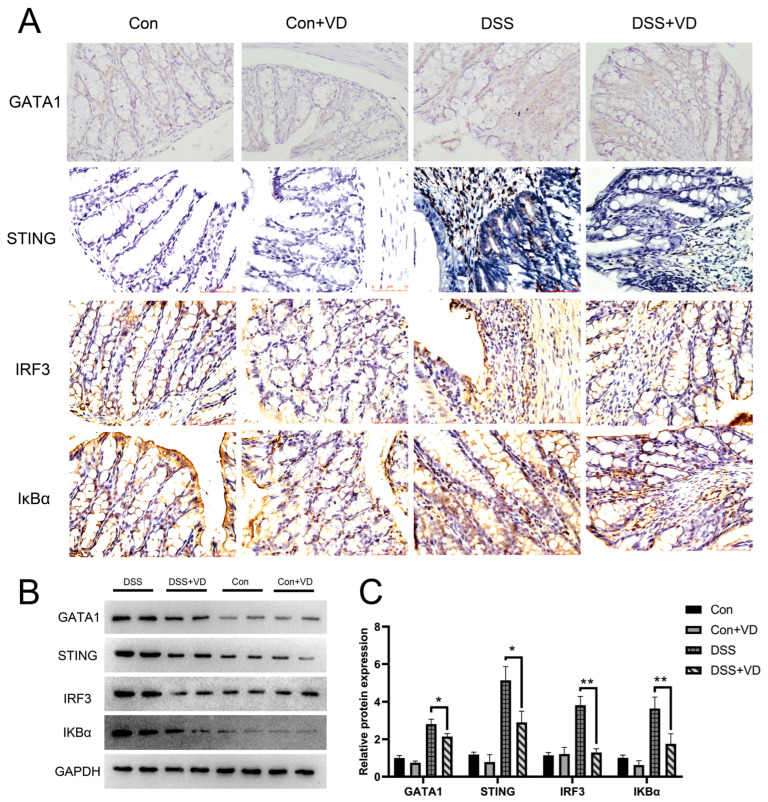
VD treatment inhibits activation of the STING pathway. (**A**) Immunohistochemical staining of GATA1, STING, IRF3 and IκBα (magnification, ×400). (**B**,**C**) Western blot and densitometric analysis of GATA1, STING, IRF3 and IκBα. * *p* < 0.05 and ** *p* < 0.01. Each group contained five mice.

## Data Availability

The data that support the findings of this study are available from the corresponding author upon reasonable request.

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
