# Peer review of "Vitamin D Modified DSS-Induced Colitis in Mice via STING Signaling Pathway"

_biology, 2025, doi:10.3390/biology14060715_

Round 1
Reviewer 1 Report
Comments and Suggestions for Authors
In this study, the authors investigated the potential therapeutic effects of vitamin D on inflammatory bowel disease (IBD) and its underlying mechanism. They concluded that vitamin D can alleviate dextran sulfate sodium salt (DSS)-induced colitis in mice by preserving the intestinal barrier’s integrity and gut microbiota, possibly mediated through STING pathway suppression. Although this study is potentially interesting and can help expand our understanding of this field, I have some critical concerns and suggestions, as described below.
Major point:
- The post hoc ANOVA test must be noted in line 155. The Mann-Whitny U test is not appropriate for multiple comparisons. The authors must repeat the statistical analysis using the appropriate tests and methods.
- Please describe briefly how to evaluate the disease activity index (line 93) and how to collect blood samples from mice (line 97).
- Unfortunately, I could not find the supplementary materials mentioned in the methods (line 141).
- I believe that “Alizarin blue” should be “Alcian blue” (line 181). Please correct it.
- The authors concluded that vitamin D improved the intestinal barrier integrity because vitamin D significantly increased ZO-1 expression. However, this is not direct evidence for the conclusion. I strongly recommend that the authors investigate the effects of vitamin D on the intestinal barrier function by measuring epithelial barrier permeability with FITC-dextran (Li et al. J Vis Exp. 2018 19;(140):57032. doi: 10.3791/57032).
- The expression levels of GATA-1 must be investigated by western blotting as well as STING, IRF3, and Iκβα. Alternatively, a summary graph of the GATA-1-expressing area in the immunostaining images must be shown like ZO-1 in Figure 1E.
- Please add appropriate references to the first sentence of the second paragraph in the discussion (lines 260–262).
Minor point:
- Please consider changing the title of Figure 2 to “Vitamin D ameliorated changes in gut microbiota in the mice with DSS-induced colitis.” The subtitle of 3.2 (line 197) should be revised as well.
- Add information about the assay kits used for ELISA of SOD and IFN-γ (line 135).
- Clarify from which group of mice each colonic specimen was isolated in Figure 1C.
- Insert a space between “(OTUs)” and “among” (line 203).
Author Response
Point 1: The post hoc ANOVA test must be noted in line 155. The Mann-Whitny U test is not appropriate for multiple comparisons.
Response: Thank you for pointing this out, we agree with this comment, Therefore, we used Tukey HSD for the post hoc ANOVA test and obtained the same statistical results. Additionally, we used the Kruskal-Wallis H method for non-parametric testing. This change can be found in line 166-167.
Point 2: describe briefly how to evaluate the disease activity index (line 93) and how to collect blood samples from mice (line 97).
Response: Thank you for pointing this out, we agree with this comment, Therefore, we have supplemented the evaluation method of the disease activity index, and the method for collecting mouse blood. DAI: weight loss (0–4, 0 to 15 % loss), stool consistency (0 points for normal, 2 points for loose stool, and 4 points for diarrhea) and hematochezia (0 points for normal, 2 points for posi-tive occult blood, and 4 points for overt hemorrhage. Blood was collected from Saphenous vein. This change can be found in line 99-102, 106.
Point 3: supplementary materials in 141 line were not found.
Response: Thank you for pointing this out, Due to file size limitations, we will re-upload a new supplementary file.
Point 4: “Alizarin blue” should be “Alcian blue” (line 181).
Response: Thank you for pointing this out, we agree with this comment, Therefore, we have made corrections. This change can be found in line 196.
Point 5: Reviewer pointed that ZO-1 expression not direct evidence for vitamin D improved the intestinal barrier integrity, the FITC-dextran should be used.
Response: Thank you for pointing this out, we agree with this comment. Our research does have limitations, and we greatly appreciate your suggestions. FITC-dextran can very intuitively help us study intestinal barrier integrity, but since we do not have enough time and animals to repeat this animal experiments, we will use this method in future research. Thank you again for your guidance and help!
Point 6: The expression levels of GATA-1 must be investigated.
Response: Thank you for pointing this out, we agree with this comment, Therefore, we have added the western blotting results for GATA-1, as well as a summary image of the GATA-1 expression areas in the immunostaining images. This change can be found in Fig 4.
Point 7: Please add appropriate references in the discussion (lines 260–262).
Response: Thank you for pointing this out, we agree with this comment, Therefore, we have added reference to make it understood. This change can be found in Ref 26, line 405-406.
Reviewer 2 Report
Comments and Suggestions for Authors
The authors used a well-characterized animal colitis model to examine the potential protective effect of vitamin D administration. They proved with different techniques (immunohistochemistry, gene expression analysis, Western blotting, measurement of secreted molecules, histological examination of colon) the benefits of vitamin D treatment. They could prove many advantages of the effect of vitamin D concerning the gene expression changes, inflammatory components, cell signaling pathways, gut microbiota composition.
The methods used are correct, the descriptions need polishing. The introduction and discussion need some adjustments. Quality of figures are not always proper. Cited literature is useful.
In detail: In the introduction the mentioned antimicrobial peptides should be shown with some examples. The STING pathway needs some more description. There are molecules which are examined in the experiments, but there is no explanation of them or their function (GATA-1, mucin-2, ZO-1, Nrf-2, Keap 1, IκBα – for example). Materials and methods – 2.3. A few of the examined genes are missing in the list (Nrf -2 for example). 2.4. Alcian blue staining protocol and the reason to use it is missing. 2.5. Again a few components are missing (Zo-1 fer example). IKK beta is not in the figures.
Results and figures: In Fig. 1 F – vitamin D administration alone increased Mucin-2 expression. What is the reason for this? In ln 177 – serum SOD level is not changing similarly to previously examined components. Fig 1 is hard to see. In Fig 1.A vitamin D injection times should be indicated also. What was the reason that only IFN-beta serum level was measured among the secreted molecules? In Fig 4 legend B is a Western blot, and nothing is written about C. In Fig 2 the order of the experimental groups is different than in other figures (starting with DSS not with control).
Discussion: STING pathway description is missing. Also, there should be a few words about the possible interaction of microbiota and colitis.
Author Response
Comments 1: the mentioned antimicrobial peptides should be shown with some examples.
Response 1: Thank you for pointing this out, we agree with this comment, Therefore, we have added some antimicrobial peptides influenced by vitamin D, such as cathelicidin and β2-defensins. This change can be found in line 53.
Comments 2: The STING pathway needs some more description.
Response 2: Thank you for pointing this out, we agree with this comment, Therefore, in the introduction section, we added some descriptions about the STING pathway. This change can be found in line 65-67.
Comments 3: Molecules examined in the experiments have no no explanation of them or their function.
Response 3: Thank you for pointing this out, we agree with this comment, Therefore, function of GATA-1、mucin-2、ZO-1、Nrf-2、Keap 1、IκBα were complemented. This change can be found in line 263-265, 191-193, 199-202,185-186, 257-258.
Comments 4: Materials and methods
Response 4: Thank you for pointing this out, we agree with this comment, Therefore, Genes such as nrf2, keap1, and mucin2 have been supplemented; Supplemented the relevant information on Alcian blue staining in section 2.4 and corrected the information on antibodies in section 2.5. This change can be found in line 116-119.
Comments 5: What is the reason for vitamin D administration alone increased Mucin-2 expression
Response 5: Thank you for pointing this out. We have not conducted an in-depth study on the occurrence of this phenomenon. In other studies, vitamin D may affect the metabolic products of gut bacteria, such as butyrate, which has been shown to increase the transcription levels of MUC2 mucin (doi: 10.1152/ajpgi.00219.2004). This may help explain the phenomenon.
Comments 6: In ln 177 – serum SOD level is not changing similarly to previously examined components.
Response 6: An analysis of SOD levels serves as an indicator of oxidative stress, aligning with the functional role of the Keap1/Nrf2.
Comments 7: Questions about Fig 1.
Response 7: Thank you for pointing this out, we agree with this comment, Therefore, the size of the pathological images in Figure 1 has been adjusted, and the injection time of vitamin D has been noted in Fig 1A.
Comments 8: What was the reason that only IFN-beta serum level was measured among the secreted molecules?
Response 8: Thank you for pointing this out, IFN-beta is a type of type-1 interferon, and its expression level increases after the activation of the cGAS/STING pathway. Therefore, we measured the levels of IFN-beta to assess the activation status of the cGAS/STING pathway in each group.
Comments 9: In Fig 2 the order of the experimental groups is different than in other figures (starting with DSS not with control).
Response 9: Due to the grouping order starting incorrectly from the DSS group during the initial gut microbiota analysis, there are some differences in the result graphs. I apologize for any confusion this may have caused you.
Comments 10: STING pathway description is missing. Also, there should be a few words about the possible interaction of microbiota and colitis.
Response 10: Thank you for pointing this out, we agree with this comment, Therefore, we have added some discussions related to gut microbiota and IBD, and linked them to STING. This change can be found in line 299-307.
Reviewer 3 Report
Comments and Suggestions for Authors
Thank you for the opportunity to review such an interesting research!
I would suggest describing the results in more detail, as for example the immunoblots could benefit from a short description of what they show, so the less experienced readers can also learn from your vast experience.
I have only one additional comment: the conclusions section is missing. I would strongly suggest adding it, to better highlight the value of the results you have obtained.
Author Response
Comments1: I would suggest describing the results in more detail, as for example the immunoblots could benefit from a short description of what they show, so the less experienced readers can also learn from your vast experience.
Response1: We have added some explanations for certain proteins or factors involved in the research, and we hope this will be helpful. Such as line 185-186,191-193,199-202,257-258, 263-265.
Comments 2: the conclusions section is missing
Response 2: We have supplemented the conclusion section in line 335-338
Round 2
Reviewer 1 Report
Comments and Suggestions for Authors
Thank you for your response and for addressing some of my earlier comments. Although the manuscript has been revised, several important issues remain unresolved. Please carefully consider the following points in your next revision:
Major Comments:
- Some of the minor points I raised during the initial round of review have not been addressed. These may have been inadvertently overlooked. I strongly recommend that the authors revisit the earlier reviewer comments to ensure that all points are thoroughly responded to and appropriately incorporated into the manuscript.
- I suggest that the authors briefly discuss the limitations of the study, particularly regarding experimental reproducibility and methodological constraints. For instance, the use of FITC-dextran is a valuable method to assess intestinal barrier integrity. However, the authors mentioned their inability to repeat certain animal experiments due to time and resource limitations. While this is understandable, it should be explicitly acknowledged in the Discussion section to provide transparency and guide future studies.
- There appears to be a discrepancy between the immunostaining results in Figure 4A, which suggest low-level GATA1 expression in control and control + vitamin D groups, and the western blot data in Figures 4B and 4C, which indicate no expression of GATA1 in these groups. This inconsistency should be clearly explained in the text. Additionally, I request that the original images of western blots for GATA1 be provided as supplementary materials to support the integrity of the data.
- The legend for Figure 4B appears to be incorrect. The legend for Figure 4C is missing entirely. The label “B” in Figure 4B is partially cut off. Please revise the figure legends and correct the labeling to ensure clarity and consistency with the figure content.
Author Response
Point 1: Some of the minor points I raised during the initial round of review have not been addressed.
Response: Thank you for pointing this out again, We have proofread and made changes. The title of Figure 2 and subtitle of 3.2 had been changed to "Vitamin D ameliorated changes in gut microbiota in the mice with DSS-induced colitis", you can find them in line 229 and line 248.The information about the assay kits used for ELISA of SOD and IFN-γ had been added ,you can find them in line 159-160. Groups of mice colonic specimen have been isolated in Figure 1C.We Inserted a space between “(OTUs)” and “among” (line 203), you can find them in line 234.
Point 2: it should be explicitly acknowledged in the Discussion section to provide transparency and guide future studies.
Response: Thank you for pointing this out again, We have proofread and made changes. In line 343-347 ,We pointed out the shortcomings of the methods used in the research process and will attempt to use this method in future studies. Thank you again for your guidance.
Point 3: The immunostaining results in Figure 4A suggested low-level GATA1 expression in control and control + vitamin D groups, and the western blot data in Figures 4B and 4C, which indicate no expression of GATA1 in these groups.
Response: Thank you for pointing this. We re-extracted the proteins from the intestinal mucosa and performed Western blotting experiments. Following an adjustment in the exposure duration, we were able to obtain results, you can find it in Figure 4A. The original images of western blots for GATA1 has been provided in supplementary materials. We appreciate your guidance in this process.
Point 4: The legend for Figure 4B appears to be incorrect. The legend for Figure 4C is missing entirely. The label “B” in Figure 4B is partially cut off.
Response: Thank you for pointing this. We have proofread and made changes.
Reviewer 2 Report
Comments and Suggestions for Authors
I have read the answers and the modified manuscript, and I agree with the changes made by the authors. I accept the manuscript for publication in this form.
Author Response
Thank you again for your guidance on our work.